# Probe Sequencing Analysis of Regenerating Lizard Tails Indicates Crosstalk Among Osteoclasts, Epidermal Cells, and Fibroblasts

**DOI:** 10.3390/jdb13020015

**Published:** 2025-05-03

**Authors:** Darian J. Gamble, Samantha Lopez, Melody Yazdi, Toni Castro-Torres, Thomas P. Lozito

**Affiliations:** 1Department of Stem Cell Biology and Regenerative Medicine, Keck School of Medicine, University of Southern California, 1425 San Pablo St., Los Angeles, CA 90033, USA; djgamble@usc.edu; 2Department of Orthopedic Surgery, Keck School of Medicine, University of Southern California, 1540 Alcazar St., Los Angeles, CA 90089, USA; lopezsamantha0427@gmail.com (S.L.); myazdi@usc.edu (M.Y.); castroto@usc.edu (T.C.-T.)

**Keywords:** blastema, regeneration, osteoclasts, zoledronic acid, lizard tail regrowth

## Abstract

Lizards are distinguished as the only amniotes, and closest relatives of mammals, capable of multilineage epimorphic regeneration. Tail blastemas of green anole lizards (*Anolis carolinensis*) consist of *col3a1^+^* fibroblastic connective tissue cells enclosed in *krt5^+^* wound epidermis (WE), both of which are required for regeneration. Blastema and WE formation are known to be closely associated with phagocytic cell populations, including macrophages and osteoclasts. However, it remains unclear what specific phagocytic cell types are required to stimulate regeneration. Here, we explicitly assess the roles of osteoclast activity during blastema and WE formation in regenerating lizard tails. First, probe sequencing was performed at regenerative timepoints on fibroblasts isolated based on *col3a1* expression toward establishing pathways involved in stimulating blastema formation and subsequent tail regrowth. Next, treatments with osteoclast inhibitor zoledronic acid (ZA) were used to assess the roles of osteoclast activity in lizard tail regeneration and fibroblast signaling. ZA treatment stunted lizard tail regrowth, suggesting osteoclast activity was required for blastema formation and regeneration. Transcriptomic profiling of fibroblasts isolated from ZA-treated and control lizards linked inhibition of osteoclast activity with limitations in fibroblasts to form pro-regenerative extracellular matrix and support WE formation. These results suggest that crosstalk between osteoclasts and fibroblasts regulates blastema and WE formation during lizard tail regeneration.

## 1. Introduction

Epimorphic appendage regrowth is characterized by the formation of blastemas, regenerative structures made up of proliferating cells that drive the regrowth of replacement tissues [1]. Epimorphic regeneration is widespread among anamniotes. For example, all known urodele species are capable of fully regenerating both limbs and tails [2]. In contrast, blastema-based regeneration is rarer and more limited among amniotes [3,4,5]. Furthermore, amniote blastemas tend to exhibit limited differentiation capacities and fidelities compared to those of anamniotes [6]. For example, mouse digit tip blastema differentiation is limited to regenerated bone [3]. Interestingly, lizards are distinguished as the only amniotes, and closest relatives of mammals, capable of epimorphic regeneration involving tissue types originating from all three germ layers [7]. Therefore, lizards represent informative model organisms toward bridging gaps in blastema formation and differentiation potentials among amniotes, including humans.

Structurally, tail blastemas of lizards like the green anole (*Anolis carolinensis*) consist of collagen type III (*col3a1^+^*) fibroblastic connective tissue cell (FCTC) populations enclosed in specialized keratin-5 (*krt5^+^*) wound epidermis (WE) [7]. These structures develop in a coordinated, interdependent manner. For example, disruption of either FCTC proliferation or WE formation prevents blastema development and leads to fibrosis [8]. Major milestones during *A. carolinensis* blastema development can be grouped into three phases based on specific characteristics exhibited by either FCTC or WE compartments. In original tails, the epidermis is associated with large, organized scales and homeostatic FCTCs line periosteum, perichondrium, dermis, and other connective tissues. Upon injury or amputation, damaged tail tissues trigger immune responses that peak around day 7 (D7). The first major regenerative phase is the inflammation phase, characterized by tail stump histolysis in which cathepsin-K (*ctsk^+^*) osteoclasts and other phagocytic cells enter injury sites and eliminate dead tissue and debris [7]. Wound closure also occurs in this phase, generating *krt5^+^* WE that forms under scabs [7]. The blastema phase occurs at day 14 (D14) and is designated by the formation of masses of stem-like mesenchymal cells. *A. carolinensis* blastema cells are derived from *col3a1^+^* fibroblasts, which eventually differentiate into most of the mesenchyme-derived tissue populations within regenerated tails [7]. The last phase involves terminal regeneration and occurs at day 28 (D28) with reconstitution of most of the tail tissues. *A. carolinensis* tail regrowth is characterized as imperfect due to differences in some regenerated tissue types compared to originals. For example, skin neogenesis is disorganized in regenerated tails with smaller, irregular scales that lack chromatophores. Furthermore, segmented ossified vertebrae of original tails are replaced with unsegmented cartilage tubes [9,10].

Blastema FCTCs are derived from resting interstitial fibroblasts liberated through histolysis of stump tissues 0–7 days post amputation (DPA). Histolysis is supported by phagocyte populations, particularly *ctsk*^+^ osteoclasts, which invade amputated tail stumps. Indeed, phagocytic cell populations have previously been shown to be vital to the regenerative process in lizards and other species capable of epimorphic regeneration [7,8,11,12,13]. Untargeted depletion of phagocytic cells via clodronate liposomes has been shown to abrogate histolysis and prevent blastema and normal WE formation [8]. Following phagocyte depletion, the same FCTC populations that would contribute to blastemas formed scars instead [8]. Taken together, these studies suggested that osteoclasts and other phagocytes contribute pro-regenerative and anti-fibrotic signals independent of histolytic activities.

Studies involving untargeted phagocyte depletion via clodronate liposomes are unsuitable for testing the specific roles of osteoclasts in epimorphic regeneration. Zoledronic acid (ZA) is an anti-osteoporosis medication that belongs to a class of drugs known as bisphosphonates [14]. In mammals, bisphosphonates inhibit bone loss and remodeling by encouraging osteoclast apoptosis [14]. ZA has been shown to decrease osteoclast activity, prevent bone resorption, and significantly reduce blastema size in the axolotl salamander [15]. Currently, the roles of osteoclasts in stimulating blastema and WE formation in lizards have not been studied.

To investigate the role of lizard osteoclasts in establishing a pro-regenerative niche for *col3a1^+^* blastema fibroblasts and *krt5^+^* WE cells, we employed probe sequencing technology to isolate and profile cell populations over time in the presence and absence of osteoclast activity [16]. Hybridization chain reaction fluorescence in situ hybridization (FISH) was used to label and isolate *col3a1^+^* fibroblasts and *krt5^+^* epidermal cells to perform bulk RNA-sequencing. With this technology, we show that osteoclasts promote a pro-regenerative, anti-fibrotic program in blastema fibroblasts.

## 2. Materials and Methods

### 2.1. Animal Ethics Statement

Husbandry and experimental use of green anole lizards (*Anolis carolinensis*) were conducted per guidelines of the Institutional Animal Care and Use Committee at the University of Southern California (protocol 20992).

### 2.2. Tail Amputations

Lizard tail tissue collection involved two amputations per animal. First amputations removed original tails and created wounds. Lizard tails were anesthetized with 10 s ethyl chloride sprays and amputated with sterile scalpel blades, leaving 1/3rd of original tails remaining. To collect tissues for histological analysis and cell isolations, lizards were euthanized, and second amputations were performed 5 mm distal to the original amputation planes, typically 14 or 28 days post amputation (DPA). Tail samples were collected in Hank’s balanced salt solution (HBSS) until further processing was required. Lizard euthanasia was performed in a manner consistent with the recommendations of the American Veterinary Medical Association (AVMA) Guidelines according to the protocol: (1) inject 25–50 µL of pH-neutralized 1% MS222 into the coelom. (2) After loss of righting reflex, inject 25–50 µL 50% (*v*/*v*) MS222 into the coelom. (3) Decapitation. (4) Double pithing of brains/spinal cords.

### 2.3. Lizard Vendor and Husbandry

*A. carolinensis* lizards were commercially obtained from USC’s institutionally approved vendor Carolina Biological Supply Company (Burlington, NC, USA). Lizards were housed in metal mesh cages, with 12 h light/12 h dark schedules using combined 100 W UVA/UVB bulbs. The vivarium was maintained at 24–26 °C during light hours and 18.5–21 °C during dark hours. Cages were misted with water five times per week, and lizards were fed a diet of ½ inch crickets supplemented with calcium. All lizards used in the study were 9–12 months old. Male and female anoles were used in equal numbers.

### 2.4. Tail Dissociation

After amputation, tails samples were placed on ice in HBSS solution. Scales and epidermis were peeled from tail samples, scored, and placed into collection tubes containing HBSS. Remaining tissue was minced and placed into the same collection tubes. Minced and scored tissue was transferred to dissociation solution containing 12 mL of HBSS, 18 mL of 0.25% trypsin (15050-057, Gibco, Waltham, MA, USA), and 30 mg of collagenase type 2 (LS004176, Worthington-Biochemical, Freehold, NJ, USA) and placed onto a heated shaker for 75 min at 37 °C. Reactions were quenched with 5 mL of FBS and filtered into 50 mL conical tubes using 40-micron strainers, and cells were pelleted and split into aliquots for FISH.

### 2.5. Histology

Lizard tail samples were fixed overnight in 10% neutral buffered formalin (HT501128, Sigma, St. Louis, MO, USA) and placed into decalcification solution (14–20% ethylenediaminetetraacetic acid (EDTA), pH 7.2) for 1 week. Samples were then equilibrated with a sucrose gradient prior to snap freezing in Optimal Cutting Temperature Compound (OCT) with isopentane and dry ice. Frozen samples were sectioned at a thickness of 16 µm using a Leica CM1860 cryostat.

### 2.6. Fluorescence In Situ Hybridization (FISH) on Tissue Sections

FISH was performed on lizard tail histological samples using the hybridization chain reaction (HCR) protocol for fixed frozen samples, MI-Protocol-RNAFISH-Fresh Fixed/Frozen Tissue Revision No.4. Tissues were fixed by immersing slides in ice-cold 4% paraformaldehyde (PFA) for 15 min at 4 °C, followed by an ethanol gradient at room temperature. A total of 200 µL of 10 µg/mL proteinase K solution was added to each sample, and slides were incubated in a humidified chamber for 10 min at 37 °C. After washing with PBS, 200 µL of probe hybridization buffer was added to each sample. Slides were pre-hybridized with buffer for 10 min inside a humidified chamber at 37 °C.

To prepare the probe solution, 20 µL of probe was added to 80 µL of probe hybridization buffer at 37 °C. Probes were commercially generated and supplied by Molecular Instruments, targeting *col3a1* and *krt5* mRNA. Each probe library contained forty unique 20 bp oligos hairpins targeting the mRNA of interest. Pre-hybridization solution was removed, and 100 µL of probe solution was added on top of each sample. Coverslips were placed on samples, and slides were incubated overnight (>12 h) in a 37 °C humidified chamber.

Coverslips were gently removed, and excess probes were washed by incubating slides at 37 °C in the following solutions containing saline-sodium citrate (J60839.K7, Thermo, Waltham, MA, USA) with 0.01% Tween-20 (SSC-T): (a) 75% probe wash buffer/25% 5× SSCT for 15 min, (b) 50% probe wash buffer/50% 5× SSCT for 15 min, (c) 25% probe wash buffer/75% 5× SSCT for 15 min, and (d) 100% 5× SSCT for 15 min. Slides were then immersed in 5× SSCT for 5 min at room temperature.

A total of 200 µL of amplification buffer was added on top of each sample. Slides were pre-amplified in a humidified chamber for 30 min at room temperature. Separately, 6 pmol of hairpin h1 and 6 pmol of hairpin h2 were prepared by snap cooling 2 µL of 3 µM stock per sample (heating at 95 °C for 90 s and cooling to room temperature in a dark drawer for 30 min). Hairpins were commercially available fluorophore-ligated oligos produced by Molecular Instruments (Los Angeles, CA, USA), with Alexafluor647 and Alexafluor594. Hairpin solution was prepared by adding snap-cooled h1 and h2 hairpins to 100 µL of amplification buffer at room temperature.

Pre-amplification solution was removed, and 100 µL of hairpin solution was added on top of each sample. Coverslips were placed on samples, and slides were incubated overnight (>12 h) in a dark humidified chamber at room temperature. Coverslips were gently removed, and excess hairpins were washed by incubating slides in 5× SSCT at room temperature for (a) 2 × 30 min and (b) 1 × 5 min. Samples were counterstained with DAPI (320858, Advanced Cell Diagnostics, Newark, CA, USA) for 90 s, and coverslips were mounted with Prolong Gold Antifade Media (P36930, Invitrogen, Waltham, MA, USA).

### 2.7. Fluorescence In Situ Hybridization (FISH) on Cells in Suspension

Cells were fixed for 1 h at room temperature, washed 4 times with PBST, and re-suspended in cold 70% ethanol. Cells were stored at 4 °C overnight before use. The desired number (0.5–1 × 10^6^) of fixed cells was transferred into 1.5 mL tubes, washed twice with 500 µL of PBST, re-suspended in 400 µL of probe hybridization buffer, and pre-hybridized for 30 min at 37 °C. Meanwhile, probe solutions were prepared by adding 20 µL of each probe set to 80 µL of probe hybridization buffer pre-heated to 37 °C. Probe solution was added directly to cell samples and incubated overnight (>12 h) at 37 °C. After incubation, the sample was centrifuged for 5 min to remove probe solution. After washing four times in probe wash buffer, cells were re-suspended in 500 µL of 5× saline-sodium citrate (J60839.K7, Thermo) with 0.01% Tween-20 (SSC-T) SSCT and incubated for 5 min at room temperature. Next, cells were re-suspended in 150 µL of amplification buffer and pre-amplified for 30 min at room temperature.

In separate preparations, 15 pmol of hairpin h1 and 15 pmol of hairpin h2 were snap-cooled, and hairpin solutions were prepared by adding h1 and h2 hairpins to 100 µL of amplification buffer at room temperature. Hairpin solution was added directly to each sample to achieve final hairpin concentrations of 60 nM. The sample was incubated overnight (>12 h) in the dark at room temperature.

Finally, cells were washed five times with 500 µL of 5× SSCT, counterstained with DAPI (320858, Advanced Cell Diagnostics), and filtered before flow cytometry analysis.

### 2.8. Flow Cytometry

Flow cytometry was performed using the BD FACSAria II cell sorter equipped with Diva 8.0.2 on Windows 7. Cells were sorted based on fluorescence intensity using 355 nm (UV), 488 nm (blue), and 633 nm (red) lasers. Cells were collected in 200 µL of 0.04% BSA for downstream processing.

### 2.9. RNA Isolation

RNA extraction was performed using the RNeasy DSP FFPE Kit (73604, QIAGEN, Hilden, Germany), according to the manufacturer’s instructions. Briefly, cells were incubated in lysis buffer containing 6.25% proteinase K for 15 min at 56 °C with shaking at 600 rpm. Cells were then cooled to room temperature for 3 min and reheated to 80 °C with shaking at 600 rpm for an additional 15 min. Samples were placed on ice for 3 min and centrifuged at 20,000× *g* for 15 min. Supernatant was collected and incubated with DNase and DNase booster buffer for 15 min. Ethanol was added to supernatants and passed though Qiagen min elute spin columns. Clean up steps were performed, and RNA was eluted in RNase free water.

### 2.10. RNA Quality Control

Isolated RNA samples were quality controlled by using a Nanodrop to assess concentration and High Sensitivity RNA ScreenTape^®^ on an Agilent 2100 Bioanalyzer to assess RNA integrity. Since samples were fixed and subjected to high temperatures during RNA isolation, RIN values were highly irregular and could drop below normally acceptable values. Instead of using typical RIN values to assess the quality of RNA, the percentage value of total RNA between 200 and 10,000 nucleotides (DV200 score) was used as a metric of successful library preparation when using formalin-fixed samples. Since a DV200 score of <20% has been shown to have a low likelihood of generating successful RNA libraries, only samples that had a DV200 of over 20% were used for this study.

### 2.11. Polymerase Chain Reaction

RNA was converted to cDNA using the SuperScript VILO cDNA synthesis kit (Invitrogen REF 11754050) and normalized to a concentration of 3 nM. Each PCR reaction contained 5 µL of 2×Taq PCR Premix (Bioland Scientific Paramount, CA Cat:TP01-00), 3 µL of nuclease free water, 0.5 µL of each forward and reverse primer, and 1 µL of cDNA. Reactions were run on a BioRad T100 Thermal Cycler using a 30 s 95 °C, 30 s 55 °C, 1 min 72 °C scheme for 32 cycles. Reactions were then run on 2% agarose gel containing 2 µL of gel green nucleic acid stain for 1 h at 150 V. Gels were then imaged using a Chemidoc MP Imaging system using a 590/110, UV transilluminator. Primer sequences are located in Appendix A.

### 2.12. Library Preparation

Library prep was performed at the University of California Los Angeles Technology Center for Genomics and Bioinformatics (TCGB). The RNA sample concentration and integrity was assessed via Nanodrop and Tapes Station. High-quality samples were synthesized into cDNA libraries using the Illumina^®^ Stranded Total RNA Prep, Ligation with Ribo-Zero Plus (C 20040525, Illumina, San Diego, CA, USA) according to the kit protocol. Supplemental oligo probes to ensure depletion of *A. carolinensis* rRNA were designed using SilvaDB to find specific lizard rRNA sequences. Oligo pools were generated by Integrated DNA Technologies (IDT) and used as directed in the library preparation protocol.

### 2.13. Sequencing

Bulk- RNA sequencing was performed at the UCLA TCGB core facility using the NovaSeq X Plus platform, with paired end 2 × 50 at a depth of 40–80 million reads per sample. Raw and normalized count files were uploaded to the NCBI Gene Expression Omnibus under GSE292905.

### 2.14. RNA Differential Gene Expression Analysis

Sequenced reads were aligned to the most recent annotated version of the *A. carolinensis* genome (rAnoCar3.1.pri, NCBI) using STAR version 2.7.6a. Differential gene expression results were generated using DESeq2 version 1.45.1 with default settings. The Top 30 differentially expressed and annotated genes were selected based on log2fold change > 2 and *p*-adj value < 0.05. Genes shown in heatmaps selected for genes most likely associated with mechanisms occurring during regenerative timepoints or ZA treatment.

### 2.15. Zoledronic Acid (ZA) Administration

Zoledronic acid monohydrate (SMLL02233, Sigma) was dissolved in PBS to a concentration of 2 mg/mL. Each lizard was treated with ZA (250 µL of 4 µL/mL solution) via intraperitoneal injection three times per week until collected for analysis. Samples were immediately processed for histology.

### 2.16. EdU Labeling

Lizards were treated with 5-Ethynyl-2′-deoxyuridine (EdU) (100 µg drug/g animal) via intraperitoneal injection 4 h before tail sample collection. Collected tails were immediately processed for histology, and Click-iT EdU Imaging kits (C01337, Invitrogen) were used to detect EdU^+^ cells according to the manufacturer’s instructions. Briefly, 1 mL of 3.7% formaldehyde in PBS was added to each slide and incubated for 15 min at room temperature. Formaldehyde was then removed, and slides were washed twice with 1 mL of 3% BSA in PBS. Wash solution was discarded, and 1 mL of 0.5% Triton^®^ X-100 in PBS was added to each slide. Samples were incubated for 20 min at room temperature and washed twice with 3% BSA in PBS. To detect EdU, a Click-iT reaction cocktail using AlexaFluor488 azide (Thermo, Waltham, MA, USA) was prepared. Permeabilization buffer was removed from sections, and samples were washed twice with 1 mL of 3% BSA in PBS. Wash solution was discarded, and 0.5 mL of the Click-iT reaction cocktail was added to each slide. Samples were then incubated for 30 min at room temperature, protected from light. Afterward, the reaction cocktail was removed, and each sample was washed once with 1 mL of 3% BSA in PBS.

## 3. Results

### 3.1. Lizard Tail Regeneration Involves a Proliferative Expansion of col3a1^+^ Fibroblast and Formation of a krt5^+^ WE

As previously stated, amputated lizard tails undergo a series of stages in which dead tissues are removed, blastemas are formed, and tail regeneration occurs. Previously described *col3a1^+^* fibroblast populations have been shown to expand to form a majority of the blastema, while *krt5^+^* epidermal cell populations form the WE [7]. Using HCR FISH as well as EdU staining, we labeled and tracked the proliferation of blastema fibroblasts and WE cells over time. In original (D0) tails, *col3a1^+^* fibroblasts are closely associated with the *krt5^+^* epidermal cell population directly under scales, with little proliferation observed in either population [Figure 1A–A″]. During the histolysis phase (D7), *col3a1*^+^ fibroblasts remained closely associated with *krt5*^+^ epidermal cells. EdU^+^ proliferating cells belonging to both *col3a1^+^* and *krt5*^+^ populations were observed in close proximities of amputation planes [Figure 1B,B′]. By the blastema stage (D14), a massive expansion of *col3a1^+^*/EdU^+^ fibroblasts, as well as the establishment of a robust *krt5^+^* population, was observed. The WE appears thick, with the inclusion of a proliferative zone directly between a strongly expressing *krt5^+^* epidermal cell population and *col3a1^+^* fibroblasts [Figure 1C–C″]. At full regeneration (D28), varied stages of skin neogenesis were observed, with the highest-expressing *krt5^+^* epidermal cells localizing closest to planes of amputation. Epidermal cells remained closely associated with fibroblast clusters that highly expressed *col3a1*. Overall, fibroblast and epidermal populations decreased in proliferation with reductions in the EdU signal throughout regenerated tails. However, EdU^+^ proliferating zones were observed at distal tips of regrown tails [Figure 1D–D″].

### 3.2. Zoledronic Acid Treatment Inhibits Osteoclast Activity and Prevents Normal WE Formation

To better understand how specific phagocytic cell populations contribute to lizard tail regrowth, we selectively inhibited osteoclast activity, which has previously been reported to impede blastema formation in other animal models [15]. Osteoclast activity was inhibited using zoledronic acid (ZA) administered intraperitoneally three times per week, beginning the week prior to amputation until collection on D14 or D28 [Figure 2A]. Control lizards were treated with vehicle control (PBS). The effectiveness of ZA treatment for inhibiting lizard osteoclast activity was verified via u-CT scans of lizard tail stumps collected at D7 from ZA-treated and control lizards. Vertebrae of control tails exhibited regions of de-mineralization, indicative of osteoclast-induced ablation [8,17]. Conversely, distal tail vertebrae of ZA-treated lizards did not exhibit ablation [Appendix A].

Comparisons of tails collected from ZA-treated versus control lizards by gross anatomy at D14 indicated that osteoclast inhibition prevented blastema formation and stunted tail regrowth [Figure 2B,D]. By D28, tails of ZA-treated lizards exhibited substantial growth inhibition and uncleared distal vertebrae, indicating the interference of vertebral ablations from osteoclast inhibition. Compared to controls, the tails of ZA-treated lizards exhibited cleft outgrowths protruding from tail stumps around distal vertebral remnants and severely diminished regenerated tail lengths [Figure 2C,E]. Histological analysis of ZA-treated tails indicated distinct reductions in *col3a1^+^* FCTCs at D14 and D28 [Figure 2F–I]. At D28, *col3a1^+^* expression localized to planes of amputation, with the highest concentrations associated with uncleared vertebra [Figure 2G,I]. While *Krt5^+^* WE cells are present in both ZA and control conditions, decreases in WE thickness were detected at D14 and D28 in ZA-treated tails [Figure 2F′–I′]. By D14, the WE of control animals consisted of stratified epidermis with low-*krt5^+^*-expressing populations directly below high-expressing *krt5^+^* populations about nine cells thick [Figure 2F′]. Conversely, the WE of ZA-treated lizards exhibited thicknesses of two–three cells [Figure 2G′]. Therefore, osteoclast inhibition by ZA treatment reduced proliferation in the layer of low-expressing *krt5^+^* WE cells and reduced overall WE thickness by nearly half. Taken together, these results suggest that osteoclast activity is required to stimulate blastema development, healthy WE formation, and subsequent tail regeneration in *A. carolinensis* lizards.

### 3.3. HCR Probe-Based Cell Labeling Allows for Targeted Isolation of Specific Lizard Tail Cell Populations

To better understand how specific cell populations contribute to tail regeneration, various isolation techniques were attempted. Current methods of lizard transgenesis do not yet support fluorescence reporter gene insertion [18]. Combined with the lack of monoclonal antibodies validated for reactivity with lizard protein epitopes, these technical challenges have historically hindered isolation of specific lizard cell populations. Recently, researchers have shown that ligation of fluorescent hybridization chain reaction (HCR) probes to specific mRNA transcripts within cells allows for fluorescence-activated cell sorting (FACS) of specific cell populations [16]. Since HCR probes are specifically designed to target any mRNA transcript regardless of species, these probes represent a preferred alternative to antibody-based methods for use in non-traditional organisms. Here, we also showed that HCR probe-based isolation of lizard cells was also compatible with downstream applications such as PCR and bulk RNA-seq analysis. In validation experiments, lizard fibroblasts were sorted based on expression of collagen type I (*col1a1*) and osteopontin (*spp1*), which were chosen based on their high expression levels in both original and regenerated lizard tail tissues [7].

Original lizard tails were digested into single-cell suspensions, labeled with DAPI and cy5-HCR probes targeting either *col1a1* or *spp1*, and analyzed by flow cytometry [Figure 3A]. DAPI labeling allowed for the selection of cells and exclusion of cell debris, and cells stained with DAPI only (DAPI control) provided gating for cy5-HCR probe-treated samples [Figure 3A]. RNA was isolated from cells belonging to each of the three labeling conditions: (1) DAPI control, (2) positive cy5-HCR label, and (3) negative cy5-HCR label. Finally, RNA was analyzed by PCR using primers targeting known markers of lizard tail cell populations [7]. For example, because lizard erythrocytes are nucleated and often resistant to red blood cell lysis techniques, contamination with erythrocytes has confounded previous attempts at transcriptomic profiling of lizard tail tissues. Thus, we were eager to test the abilities of *col1a1*- and *spp1*-targeted HCR probes to exclude erythrocytes, and we included conditions utilizing primes to hemoglobin-D (*hbD*). Both *col1a1^+^* [Figure 3B] and *spp1^+^* [Figure 3C] populations expressed high levels of their specific fibroblast marker and undetectable contamination of erythrocytes. Specifically, both positively selected HCR-labeled fibroblast populations exhibited higher expression for their respective fibroblast markers compared to negative populations and unsorted DAPI controls, suggesting true enrichments of fibroblast populations. HCR-isolated fibroblast populations also exhibited lower expression of erythrocyte and immune cell markers *hbD* or *ifi30* compared to negative populations and unsorted cells. These experiments validated probe-based isolation of lizard cells as capable of isolating pure populations of cells and yielding RNA capable of downstream analysis.

### 3.4. Col3a1^+^ Lizard Fibroblast Populations Establish Pro-Regenerative Programs During Tail Regrowth

After validating the efficacies of HCR probe-based labeling for isolating specific cell populations, we used this technology to target *col3a1^+^*/*krt5^−^* fibroblasts to assess their roles in blastema formation and subsequent regeneration. Lizard tail *col3a1^+^*/*krt5^+^* populations collected at D7, D14, and D28 were profiled via bulk RNA-seq to identify genes and potential regulatory networks activated during lizard tail regrowth. Cells were isolated based on expression of *col3a1* [Figure 4A], and RNA was isolated from each population and analyzed by bulk RNA sequencing. Experiments were repeated three times (*n* = 3) for each timepoint. Sequencing results were grouped using principal component analysis [Figure 4B]. Replicates clustered together based on timepoint, except for a D14 sample, most likely due to batch effects, as this sample was sequenced before the other eight. Each timepoint was analyzed using DESeq2 version 1.45.1, and the top 30 annotated genes were investigated deeper to understand their putative role in their respective regenerative phases. A heatmap of relative Z-scores was generated for each selected gene in each regenerative phase and presented across all timepoints [Figure 4C]. As expected, D7 timepoints exhibited increases in inflammatory-associated genes, such as *il-6* and *socs3*, and in markers associated with matrix reorganization, such as *mmp9*, *adamts4*, and *sulf1*. D14 timepoints exhibited increases in epidermal growth factor receptor (*egfr*), which plays a critical role in stimulating proliferation [19] and could be responsible for the expansion of blastema fibroblasts and WE cells. D14 samples were also enriched for *foxn1,* a gene associated with skin, nail, and hair development, as well as ECM establishment [20], and could be indicative of WE formation and the beginnings of neogenesis that occurs at proximal portions of the blastema. D14 was the only timepoint group to be consistently enriched for unannotated genes, and we are in the process of validating and identifying potential orthologues in other animal species. D28 timepoints exhibited distinctive increases in the expression of collagens and matrix building-associated genes, such as *col2a1*, *col9a2*, and *sox10*. These genes are reported to be involved in establishing the ECM required to form cartilage [21,22,23], the primary skeletal tissue found in regenerating lizard tails. Taken together, these results indicated that RNA profiled from HCR-sorted fibroblasts yields sequencing data that corroborate previous observations involving the inflammatory, blastema, and regenerated phases of regeneration [5,7,24], thereby validating probe sequencing as a powerful research tool to analyze specific cell populations in non-traditional model organisms.

### 3.5. ZA Treatment Inhibits the Formation of Matrix-Building Pathways and Stimulates Genes Associated with Fibrotic Scarring in Fibroblasts

After validating probe sequencing methods for use in lizards and identifying upregulated pathways present during normal tail regeneration over time, we decided to investigate how fibroblasts are affected by the inhibition of osteoclast activity via ZA treatment. Lizards were treated intraperitoneally with ZA or vehicle control (PBS) the week prior to amputation and during amputated tail healing until collection at D21. Three sample tails of each condition (*n* = 3) were collected and digested into single-cell suspensions. Cells were labeled with cy5-HCR probes for *col3a1* and *krt5,* and *col3a1^+^/krt5^−^* fibroblasts were isolated via FACS [Figure 5A]. Fibroblast RNA was extracted and sequenced. Biological replicates were clustered using PCA analysis and showed similar batch effects [Figure 5B]. Samples were grouped by the top 30 differentially expressed and annotated genes, and a heatmap was generated using Z-score values for selected genes in ZA and control groups [Figure 5C]. Select gene expression patterns identified from sequencing results were validated with PCR [Figure 5D].

Fibroblasts isolated from control tails expressed extracellular matrix markers indicative of cartilage formation, including *col9a2*, *col2a1,* and *sox10* [Figure 5C,D]. Conversely, fibroblasts isolated from ZA-treated tails expressed pro-fibrotic myofibroblast markers such as *myb* [25] [Figure 5C,D]. These results suggest that inhibition of osteoclast activity interferes with fibroblast expression of matrix-building genes and potentiates a fibrotic response.

### 3.6. Proposed Mechanism of Crosstalk Among Osteoclast, Fibroblast, and Epidermal Cell Populations Supporting Blastema and WE Formation During Lizard Tail Regeneration

We propose the following mechanism for the role of osteoclast activity in regulating lizard blastema and WE formation [Figure 6]. Osteoclasts support histolysis to resorb bone and stimulate an inflammatory response at D7 following tail amputation. By D14, osteoclast activity stimulates a robust expansion of closely associated pro-regenerative *col3a1^+^* fibroblasts and *krt5^+^* epidermal cells to form blastema and WE, respectively. Proliferation of both cell populations persists through D28, when tail tissues are fully differentiated. Conversely, when osteoclast activity is inhibited by ZA treatment, both blastema and WE fail to form. We hypothesize that osteoclast inhibition leads to dysregulated fibroblast signaling characterized by reduced proliferative capabilities and impaired expression of pro-regenerative ECM. These irregular fibroblasts do not support proper formation of *krt5^+^* WE and instead contribute to scar formation.

## 4. Discussion

As amniotes, lizards are the closest relative to mammals capable of appendage regeneration and represent informative model organisms for studying the divergent healing responses between lizards and mammals. Little information is available concerning the molecular mechanisms driving epimorphic regeneration in lizards compared to urodele amphibians. Similarly, roles that specific phagocytic cell types contribute to blastema formation and subsequent regeneration remain uninvestigated. The specific osteoclast inhibitor ZA afforded opportunities for targeted investigations into the roles of osteoclast activity in regulating lizard tail regeneration. We showed that inhibition of osteoclast activity led to impaired distal vertebral ablation and shedding and subsequent dysregulation of lizard tail regrowth. Furthermore, we validated probe sequencing as an effective technique for examining transcriptomes of specific cell populations while establishing baseline gene expression profiles for blastema fibroblast populations during regeneration.

Though previous reports have identified *col3a1^+^* fibroblasts populations as the primary contributor of blastema formation, the proliferative potentials of these cells during critical stages of the regenerative process have remained unstudied [7]. Here, we are the first to report on *col3a1^+^* cells with EdU^+^ signal, showing that these blastema fibroblasts indeed proliferate to form most new tissues in regenerating lizard tails [Figure 1]. Histological analyses also confirmed that lizard blastema fibroblasts exhibited similar lineage restrictions to those reported during salamander limb regeneration and were excluded from generating new muscle or nervous tissue [7,26].

We also investigated the proliferative potential of *krt5^+^* epidermal cells during WE formation. Proliferating epidermal cells localized to the deepest WE layers adjacent to col3a1^+^ dermal fibroblasts [Figure 1]. The topic of crosstalk between these two closely associated and proliferative populations has been previously suggested in the context of skin neogenesis and apical epidermal cap formation [27,28]. From our data, fibroblasts may play multiple roles during lizard tail regeneration. For example, WE-associated fibroblasts support skin neogenesis, while proliferating interstitial fibroblasts drive WE expansion during regenerating tail elongation. Future work will further explore crosstalk between fibroblasts and epidermal cells during lizard tail regrowth.

ZA treatment was used to investigate the role of osteoclast activity during lizard tail regeneration, and we describe several negative effects of osteoclast inhibition on regeneration. Treatment with ZA prevented blastema formation, as well as severely stunted lizard tail regrowth. Interestingly, osteoblast inhibition caused dysregulation in both fibroblast and epidermal compartments. ZA treatment caused reductions in blastema fibroblast and WE cell proliferation, leading to smaller blastemas and thinner WE. Furthermore, ZA treatment affected proliferation within specific regions of regenerating lizard tails, suggesting spatial dependencies of osteoclast signaling. For example, ZA treatment caused reductions in proliferative *krt5*^+^ epidermal cells within the deepest WE layers in contact with blastema fibroblasts [Figure 2]. In urodele and Xenopus tail blastema models, nervous tissue from spinal cords connects with WE to generate signaling centers that stimulate migration of regeneration-competent cells to blastemas [29,30]. Something similar might be happening in regenerating lizard tails, in which WE maturation and thickening are inhibited due to loss of osteoclast activity, resulting in the disappearance of signaling centers, diminished proliferative capacities, and failures to regenerate.

Comparing gene expression profiles of *col3a1^+^* blastema fibroblasts isolated from amputated tails of ZA-treated versus control lizards provided additional insights into the roles of osteoclast activity in supporting blastema formation and regeneration. Specifically, we showed that osteoclast signaling promoted a pro-regenerative phenotype among tail fibroblasts, marked by the expression of anabolic ECM genes such as *col9a2*, *col11a2*, and *col2a1*. Osteoclast inhibition caused these same fibroblast populations to shift toward a fibrotic myofibroblast phenotype marked by catabolic genes such as *mmp9*. Taken together, we hypothesize that lizard osteoclasts also work to create an environment that favors fibroblast proliferation and regenerative matrix deposition over myofibroblast differentiation and scar deposition.

Probe sequencing technology has previously been used in other non-traditional organisms to circumvent the need for traditional antibody or genetic labeling techniques [16]. Here, we validate the technique for use in *A. carolinensis* lizards to isolate and compare fibroblast populations over the time course of tail regrowth and in the presence and absence of osteoclast activity. Probe-based isolation of fibroblast populations was achieved using a commercially available HCR platform. Lizard fibroblast enrichment via HCR probes was validated via PCR, and probe sequencing of HCR-sorted fibroblasts confirmed previously reported expressional trajectories during tail regrowth [7]. This study also utilized probe sequencing to profile the effects of osteoclast inhibition on fibroblast fate determination between blastema fibroblasts versus myofibroblasts. Probe sequencing will be utilized in future investigations into specific molecular interactions among osteoclasts, blastema fibroblasts, and WE cells.

## Figures and Tables

**Figure 1 jdb-13-00015-f001:**
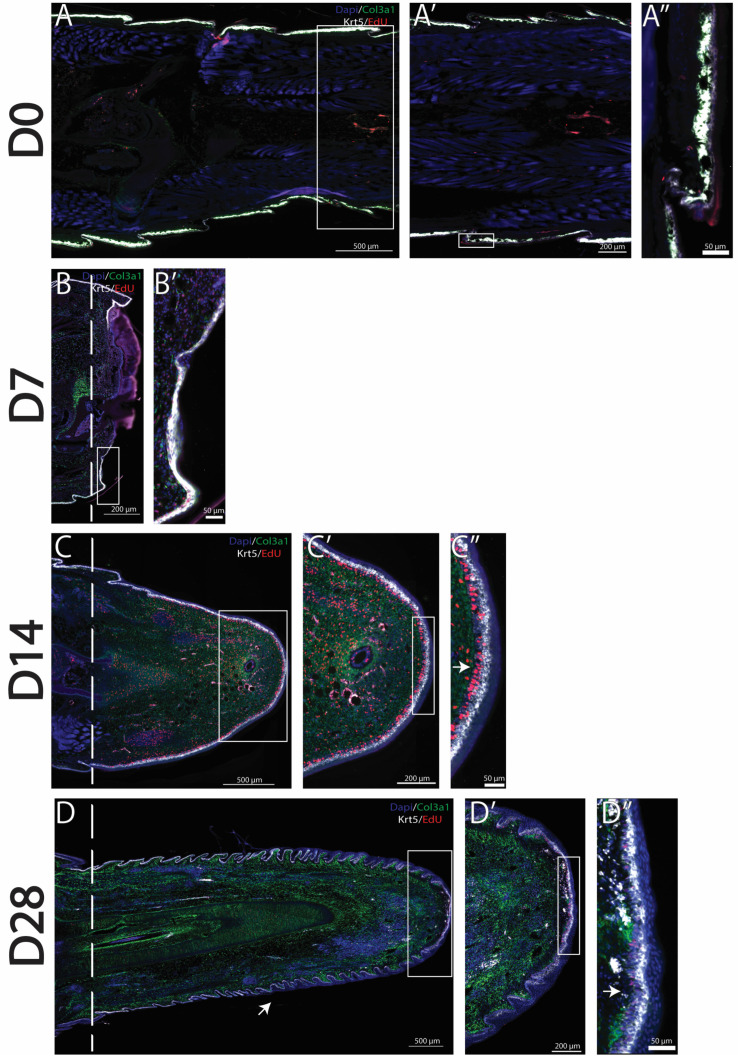
*Col3a1^+^* fibroblasts and *krt5^+^* epidermal cells proliferate to form the blastema and WE. Time course analyses of tail regeneration in green anole lizard (*Anolis carolinensis*), including (**A**) original tail (D0), (**B**) inflammation stage (D7), (**C**) blastema stage (D14), and (**D**) fully regenerated (D28) stages. Zoomed images showing the tail tip of (**A’**) original tail (D0), (**B’**) inflammation stage (D7), (**C’**) blastema stage (D14), and (**D’**) fully regenerated (D28). Sagittal tail sections are presented, and *col3a1^+^* cells are labeled in green, *krt5^+^* cells labeled in white, EdU^+^ proliferating cells labeled in red, and DAPI counterstain shown in blue. Dashed lines mark planes of amputation. White boxes indicate areas enlarged. (**A”**) shows the location of *col3a1^+^ and krt5^+^* cells in the original tail. (**C”**) Arrow marks proliferative WE zone under high-expressing *krt5^+^* cells. (**D**) Arrow marks small, unorganized regenerated scale formation. (**D”**) Arrow marks proliferative cells at distal tail tip. These findings established *col3a1*^+^ fibroblasts and *krt5*^+^ epidermal cells as populations that contribute to the canonical *A. carolinensis* regenerative process through the formation of blastema and WE.

**Figure 2 jdb-13-00015-f002:**
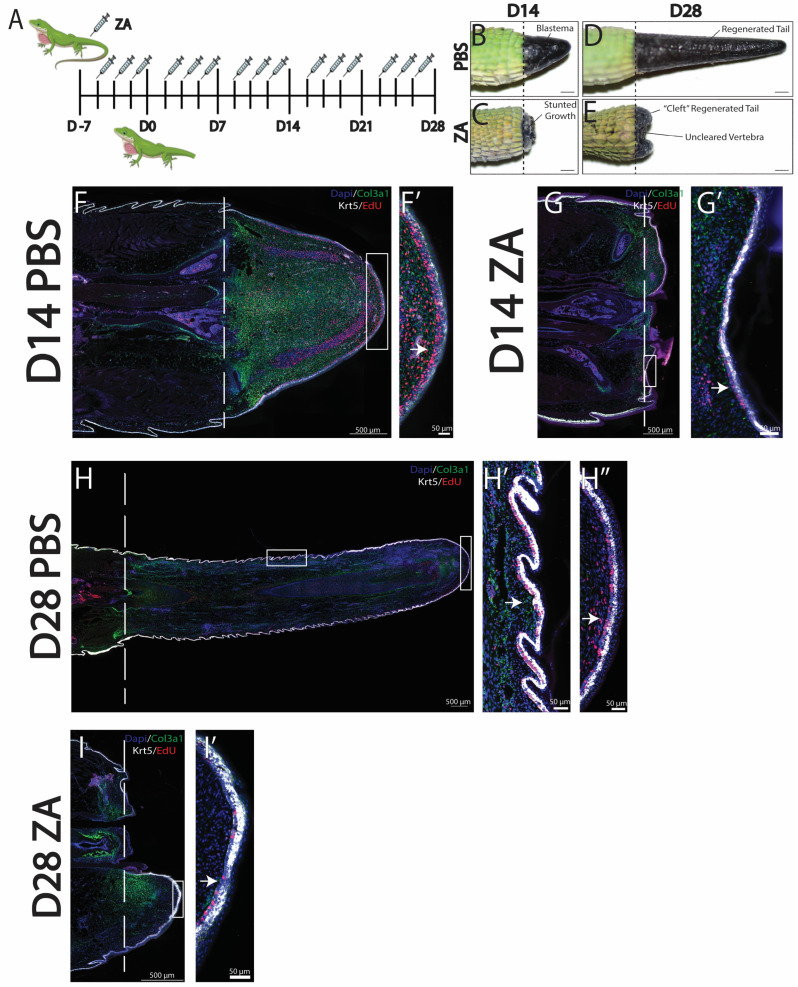
Inhibition of osteoclast activity with zoledronic acid (ZA) treatment inhibits blastema and WE formation. (**A**) Injection scheme of ZA administered 3 times a week starting 7 days prior to amputation and continuing through to D28. (**B**–**E**) Gross morphology of representative tails collected at D14 and D28 from (**B**,**C**) PBS vehicle control or (**D**,**E**) ZA. (**D**) At D14, ZA treatment caused loss of tail blastema formation and stunted regrowth. (**E**) At D28, ZA-treated tails exhibited uncleared vertebra and cleft outgrowths. (**F**–**I**) Histological analyses of representative sagittal tail sections collected at D14 and D28 from lizards treated with ZA or vehicle control. *Col3a1^+^* fibroblasts are shown in green, *krt5^+^* epidermal cells in white, EdU^+^ in red, and DAPI counterstain in blue. Dashed lines mark planes of amputation. White boxes indicate areas enlarged. (**F**,**F′**) Tails collected at D14 from control lizards exhibited *col3a1^+^* blastemas surrounded by thick bands of proliferating *krt5^+^* WE, indicated by a white arrow. (**G**,**G′**) Tails collected at D14 from ZA-treated lizards exhibited reduced *col3a1^+^* fibroblast populations and thin *krt5^+^* WE with strong reductions in proliferative signals, indicated by arrows. (**H**,**H′**,**H″**) Tails collected at D28 from control lizards exhibited expansions of *col3a1^+^* fibroblasts and patterned *krt5^+^* epidermis undergoing skin neogenesis, with high levels of proliferation (**H′**) under scales (**H″**) and at the distal tip, indicated by arrows. (**I**,**I′**) Tails collected at D28 from ZA-treated lizards exhibited *col3a1^+^* fibroblasts and *krt5^+^* WE with reduced proliferation, indicated by arrows.

**Figure 3 jdb-13-00015-f003:**
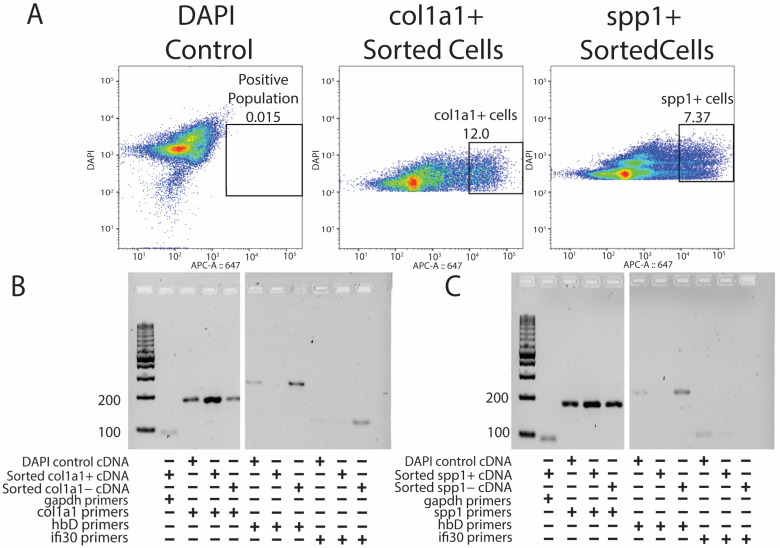
Probe sequencing allows for the enrichment of pure fibroblast populations without erythrocyte and immune cell contamination. (**A**) DAPI control (left), HCR-sorted *col1a1^+^* (middle), and HCR-sorted *spp1^+^* (right) FACS results indicating the percentage of positive cells collected. (**B**) PCR results showing the presence of *col1a1^+^* gene expression amplified from HCR-sorted *col1a1^+^* cells without *hbD and ifi30* expression. (**C**) PCR results showing the presence of *spp1^+^* gene expression from *spp1^+^* HCR-sorted populations without *hbD and ifi30* expression.

**Figure 4 jdb-13-00015-f004:**
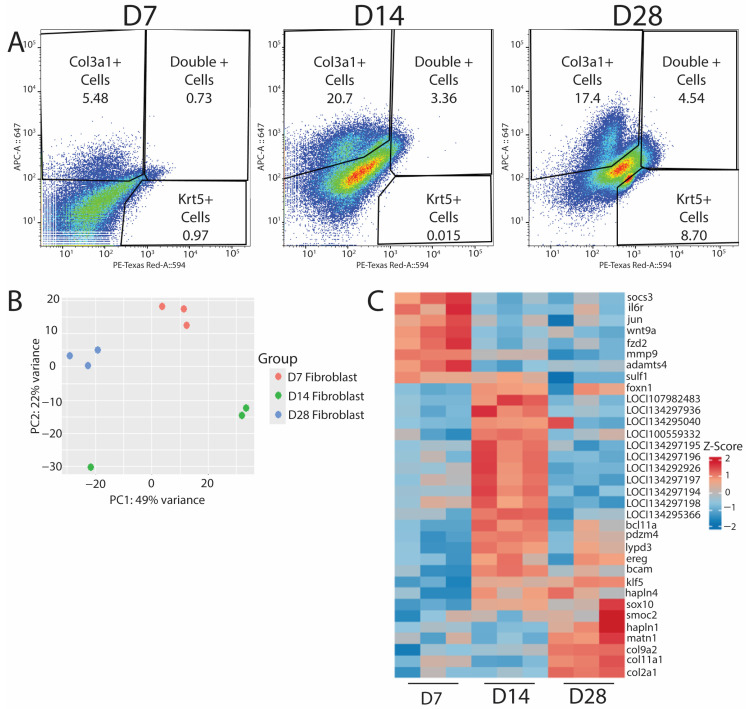
Probe sequencing of fibroblast populations over time in regenerating lizard tails exhibit transitions from immune response to matrix establishment. (**A**) D7 (left), D14 (middle), and D28 (right) FACS results indicating the percentage of *col3a1^+^* fibroblasts, *krt5^+^* epidermal cells, or double-positive cells collected for bulk RNA-seq analysis. (**B**) PCA plot of D7, D14, and D28 populations from bulk RNA-seq analysis. (**C**) Heatmap of top differentially expressed genes for D7, D14, and D28. D7 shows an upregulation of immune response genes. D14 shows an upregulation of proliferative pathways. D28 shows an increase in matrix building- and chondrogenic-associated genes.

**Figure 5 jdb-13-00015-f005:**
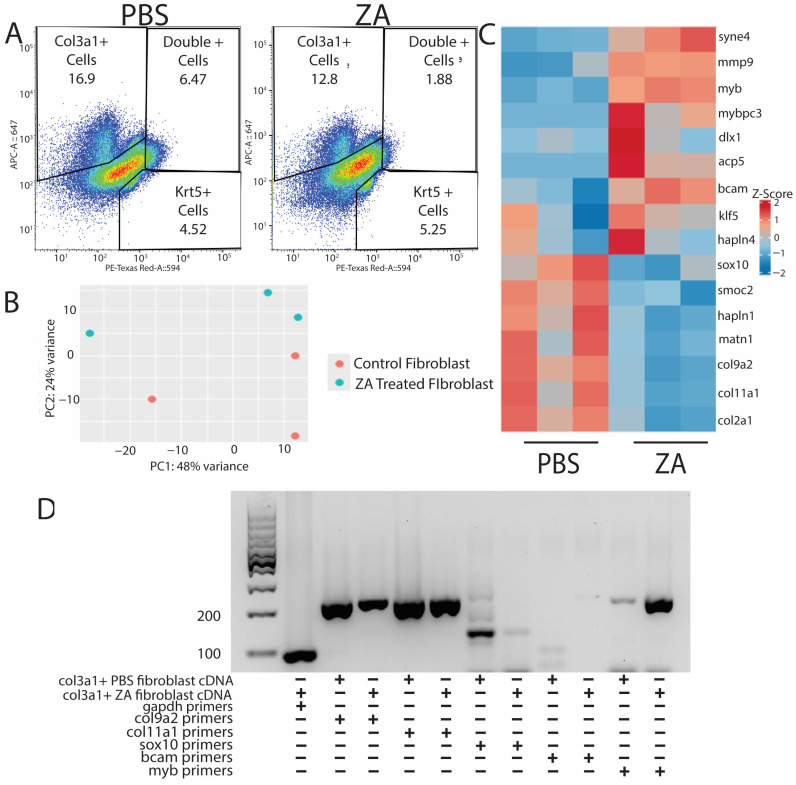
Probe sequencing of fibroblast populations from osteoclast-inhibited tails exhibits downregulation of ECM-associated genes and stimulation of genes associated with fibrotic scaring. (**A**) PBS (left) and ZA (right) *col3a1^+^* and *krt5^+^* cell populations collected via FACS from D21 tails. (**B**) PCA plot of PBS- and ZA-treated fibroblast populations from bulk RNA-seq analysis. (**C**) Heatmap of top differentially expressed genes from HCR-sorted *col3a1^+^* fibroblasts collected from lizards treated with PBS or zoledronic acid. Matrix-associated genes are highly dysregulated under osteoclast inhibition, while fibrotic scar markers are upregulated. (**D**) PCR showing expression of *col9a2*, *col11a1*, *sox10*, *bcam*, and *myb* in *col3a1^+^* fibroblasts collected at D21 from PBS- and ZA-treated lizards.

**Figure 6 jdb-13-00015-f006:**
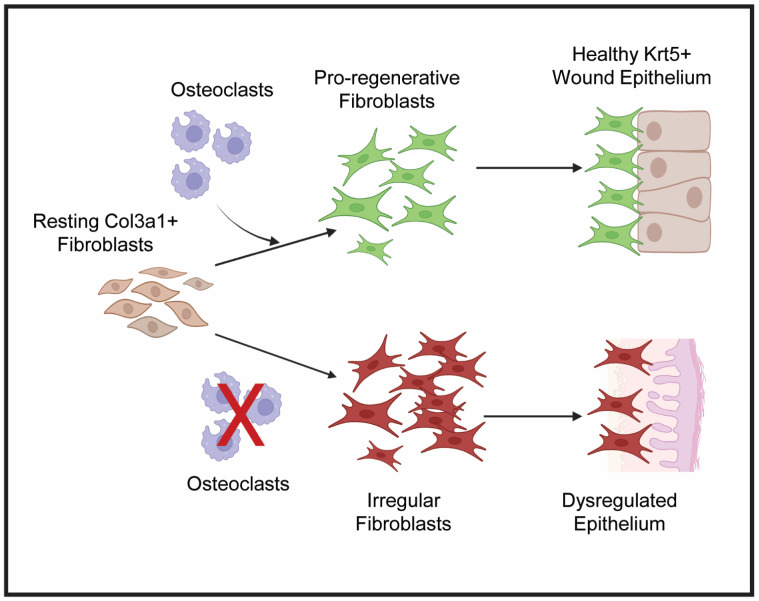
Proposed hypothesis for osteoclast-regulated crosstalk between lizard tail blastema fibroblasts and WE. Under normal conditions, osteoclast activity stimulates resting *col3a1^+^* fibroblasts to proliferate and expand to form the blastema. These pro-regenerative fibroblasts communicate with *krt5^+^* epidermal cells to form healthy, thick WE. When osteoclast activity is inhibited, symbolized by the red X, fibroblasts become irregular and fail to proliferate and establish pro-regenerative ECM. Failed fibroblast expansion and matrix establishment leads to stimulation of fibrotic genes and reduced proliferative capabilities of dysregulated WE. Created by Bio Render version 201.

## Data Availability

All sequencing data have been submitted to the NCBI Gene Expression Omnibus under GSE292905. All data will be made available upon request from the corresponding author.

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
