# Peer review of "Probe Sequencing Analysis of Regenerating Lizard Tails Indicates Crosstalk Among Osteoclasts, Epidermal Cells, and Fibroblasts"

_jdb, 2025, doi:10.3390/jdb13020015_

Round 1
Reviewer 1 Report
Comments and Suggestions for Authors
The manuscript by Gamble et al focuses on the requirement of osteoclast activity on tail regeneration in the lizard. A previous study has shown that osteoclast activity is required for skeletal absorption following amputation, which is required for proper blastema formation (Riquelme-Gusman C et al. Osteoclast-mediated resorption primes the skeleton for successful integration during axolotl limb regeneration. eLife. 2022.). Gamble et al identify possible mechanisms of osteoclasts in this process. First, they characterize the Col3a1+ fibroblasts and krt5+ epithelial cells during tail regeneration. Then they show that inhibition of osteoclast activity with zoledronic acid (ZA) inhibits blastema and wound epithelium formation following tail amputation, resulting in a lack of regeneration. Lastly, and most notably, they probe-sequencing to isolate the fibroblast populations from the blastema to determine gene expression changes during amputation-induced tail regeneration with and without ZA treatment. They can show that in the first phase, there is an upregulation of immune response genes, in the second phase there is an upregulation of proliferation genes, and in the final phase there is an increase in matrix-associated genes. When the lizard is treated with ZA, there is decreased expression of matrix-associated genes and increased expression of fibrotic genes. This suggests that osteoclast activity is required to generate pro-regenerative fibroblasts that stimulate the epithelial cells to form a wound epithelium during regeneration. Overall, the manuscript provides solid functional data for the role of osteoclast activity in regeneration, but the manuscript can be improved with some modifications to the microscopy images and verification of the sequencing data.
Major Issues:
1. Figure 1C: The zoomed in image is still too difficult to see. Please zoom in further (similar to figures 2J/K) to allow for the visualization of the proliferative zone between the fibroblasts and epithelial cells. Also, this image is missing the arrow that is supposed to be showing the proliferative zone.
2. Figure 1D: This image is also missing the arrow that shows the abnormal scale formation and group of proliferative cells at the distal tip.
3. Figure 2: How do you know that the ZA effects are specific to osteoclast inhibition? Have you verified that this inhibition is occurring in your model system? Are there other osteoclast inhibitors (other bisphosphonates or denosumab) that show similar effects to suggest that these outcomes are not due to off-target effects?
4. Figure 2: Please include a zoomed-in image of D28 (similar to Figures 2 J,K) to show the maintenance of this phenotype during regeneration.
5. Line 345: it is mentioned that not all the D28 replicates cluster together. However, from the figure it is a D14 replicate that does not cluster with the other two replicates. Is there a labeling error in the image (Figure 4B) or should the text state that it is a D14 replicate and not D28? This lack of clustering is explained as being sequenced at a different time than the other samples. Was the D28 or D14 sample sequenced separately? This discrepancy is again in lines 345, 367, and 382.
6. Figure 4: For the collection of col3a+ fibroblast cells did this include the col3a+/krt5+ or the col3a+/krt5-? From the text, it sounds like the col3a+/krt5- cells were the ones collected. If so, please specify this in the text? If this is not the case, and the col3a+/krt5+ cells were used can it be explained why this population was chosen as it may not be a pure population of fibroblasts?
7. Figure 5: Why was D21 chosen for this experiment rather than D14 or D28, which were used previously? Does this time point correspond more to a proliferative state (like D14) or ECM-producing state (like D28)? Please elaborate on this choice in experimental design.
8. Figure 5: Interesting gene expression changes are seen with the RNA-sequencing data. It would be beneficial to verify some of these changes (ECM and fibrotic scarring genes) with PCR, especially since the samples don’t cluster together.
Minor Issues:
1. Figure 1B: Is there a reason why a region behind the amputation plane is chosen to show when on the other days the distal tip of the tail is shown? Does proliferation occur at the tip of the tail or only near the amputation plane? Also, from this figure, it seems like the proliferation occurs behind the amputation plan at the top of the tail and in front of the amputation plane at the bottom of the image.
2. Line 271: Are you referring to Figure 2 D,E or figure 2 C,E instead of Figure C,D?
3. Line 275: Are you referring to Figure 2 I instead of Figure 3H?
Author Response
We greatly appreciate you taking the time to read our submission and give insightful feedback on our work. We feel as though we have addressed all comments that we have received to the best of our ability in the time allotted. We hope you are receptive to the changes we have made to our submission.
REVIEWER 1 COMMENTS
Major Issues:
- Figure 1C: The zoomed in image is still too difficult to see. Please zoom in further (similar to figures 2J/K) to allow for the visualization of the proliferative zone between the fibroblasts and epithelial cells. Also, this image is missing the arrow that is supposed to be showing the proliferative zone.
Zoomed in on Figure 1 to include proliferative, EdU+ zone for all timepoints. Removed arrows indicating proliferative zones as they are easier to see. Included arrow indicating the proliferative zone in 1C’’
- Figure 1D: This image is also missing the arrow that shows the abnormal scale formation and group of proliferative cells at the distal tip.
Included arrow showing scale formation 1D and proliferative zone 1D’.
- Figure 2: How do you know that the ZA effects are specific to osteoclast inhibition? Have you verified that this inhibition is occurring in your model system? Are there other osteoclast inhibitors (other bisphosphonates or denosumab) that show similar effects to suggest that these outcomes are not due to off-target effects?
To show that there are no indirect effects of ZA treatment we used a different bisphosponate (palmidroante) to show osteoclast inhibition was the cause of decreased proliferation and regeneration. Palmidronate caused a similar effect where the outgrowth of the blastema was reduced and portions of bone can be seen sticking out of the amputation plane. We also included micro–CT images (supplemental figure 1) showing that in ZA treatment, bone edges are more rough and appear unabsorbed compared to PBS control tails at 14 DPA. Palmidronate treatment showed a similar effect to ZA preventing blastema formation. Arrow indicates the area where the unabsorbed bone is located. Supplemental 1 at the end of paper.
- Figure 2: Please include a zoomed-in image of D28 (similar to Figures 2 J,K) to show the maintenance of this phenotype during regeneration.
Created zoomed in images of all sections to show phenotype maintenance of phenotype located in Figure 2.
- Line 345: it is mentioned that not all the D28 replicates cluster together. However, from the figure it is a D14 replicate that does not cluster with the other two replicates. Is there a labeling error in the image (Figure 4B) or should the text state that it is a D14 replicate and not D28? This lack of clustering is explained as being sequenced at a different time than the other samples. Was the D28 or D14 sample sequenced separately? This discrepancy is again in lines 345, 367, and 382.
Corrected D28 to state D14 was the batch effect outlier. D28 adjusted throughout the text be in line with figure 4. One sample from D14 was analyzed at an earlier time point to ensure sequencing was able to be performed. Cleared discrepancy from other lines in text. (lines 430-431 and 470 and 471)
- Figure 4: For the collection of col3a+ fibroblast cells did this include the col3a+/krt5+ or the col3a+/krt5-? From the text, it sounds like the col3a+/krt5- cells were the ones collected. If so, please specify this in the text? If this is not the case, and the col3a+/krt5+ cells were used can it be explained why this population was chosen as it may not be a pure population of fibroblasts?
Specified that col3a1+/krt5- cells were collected in all sections where applicable. Line 423 on.
- Figure 5: Why was D21 chosen for this experiment rather than D14 or D28, which were used previously? Does this time point correspond more to a proliferative state (like D14) or ECM-producing state (like D28)? Please elaborate on this choice in experimental design.
Figure 4 shows that between D14 and D28 fibroblasts shift from a proliferative state to a EMC-producing state. D21 represents a transition from proliferation to ECM establishment. In an attempt to better understand gene expression changes during this transition period we collected these timepoints in the presence and absence of osteoclast activity.
- Figure 5: Interesting gene expression changes are seen with the RNA-sequencing data. It would be beneficial to verify some of these changes (ECM and fibrotic scarring genes) with PCR, especially since the samples don’t cluster together.
We verified genes from figure 5 RNA-sequencing data. line 476-481.
“Fibroblasts isolated from control tails expressed extracellular matrix markers indicative of cartilage formation including col9a2, col2a1, and sox10 [Figure 5C, D]. Conversely, fibroblasts isolated from ZA-treated tails expressed pro-fibrotic myofibroblast markers such as myb [25] [Figure 5C, D]. These results suggest that inhibition of osteoclast activity interferes with fibroblast expression of matrix-building genes and potentiates a fibrotic response.”
Minor Issues:
- Figure 1B: Is there a reason why a region behind the amputation plane is chosen to show when on the other days the distal tip of the tail is shown? Does proliferation occur at the tip of the tail or only near the amputation plane? Also, from this figure, it seems like the proliferation occurs behind the amputation plan at the top of the tail and in front of the amputation plane at the bottom of the image.
Adjusted Figure 1B and B’ to show proliferation, krt5+, and col3a1+ cells after the amputation plane. Proliferation here occurs in both fibroblast and epithelial cell populations at both the tip of the tail and the amputation place. Proliferation occurs both behind and in front of the amputation place in the epithelial cells to form the WE. We are mostly focused on proliferation occurring after the plane of amputation as part of the WE.
- Line 271: Are you referring to Figure 2 D,E or figure 2 C,E instead of Figure C,D?
Referenced figures 2B-2E at the end of all statements made regarding ZA vs PBS treated lizard’s gross histology. line 340-344.
- Line 275: Are you referring to Figure 2 I instead of Figure 3H?
Referenced figures 2B-2E at the end of all statements made regarding ZA vs PBS treated lizard’s gross histology. line 340-344.
Reviewer 2 Report
Comments and Suggestions for Authors
The article by Gamble et al. is devoted to the study of the influence of osteoclasts on the regeneration process. This is a new original study performed on a unique model of modern regenerative biology - tail regeneration in the lizard Anolis carolinensis. The authors applied an original approach of isolating a population of cells expressing a specific gene using hybridisation chain reaction and flow cytometry. The method was convincingly validated. In general, I find the idea and purpose of the article interesting and relevant, but there are a number of serious remarks:
(1) It is incorrect to write throughout the text that the described phenomena are characteristic of ‘lizards’. The data were obtained for a specific species, and it is impossible to extrapolate them to all lizards. I recommend the authors to specify the species name at the very beginning of the text, otherwise it is not clear up to the M&M's section on which species the study was carried out.
(2) Regarding markers of cell populations, it is necessary to clarify in the text whether we are talking about gene transcripts or proteins. The names of markers are variously indicated everywhere (KRT5+, Krt5+ or krt5+, etc.). If we are talking about expressed genes - the name should be indicated in italics. In addition, it should be clarified which molecules are being referred to (probably collagen type III and keratin), as this makes obvious biological sense to the reader. The same applies to all genes mentioned in the article.
(3) The article is very poor in referring to data sources. Thus, in the introduction the statements are not supported by appropriate references. For example, ‘blastema-based regeneration is rarer and more limited among amniotes’, ‘amniote blastemas tend to exhibit limited differentiation capacities and fidelities’, ‘lizard tail blastemas consist of Col3a1+ fibroblastic connective tissue cells’, etc. For example, the last statement is followed by reference #4, which fails to mention the role of Col3a1+ cells in blastema formation. This problem occurs throughout the manuscript, but is especially noticeable in the Discussion section - there is not a single reference there. Because of this, it is impossible to understand where in the Discussion section the authors' own data and where the citations are. As a result, I cannot evaluate this section objectively.
(4) The descriptions of the regeneration process are very sparse and lack structure. The reader will be completely unclear the picture of events occurring at the cellular level during regeneration. At the same time, there are papers on the morphology of tail regeneration in lizards, but they are not cited in the manuscript. For example, 10.1002/jmor.20838, 10.1002/jmor.1051140305 (and other Simpson articles).
(5) I believe that RNA-seq data should be published in the NCBI SRA in accordance with the principles of transparency and reproducibility. The authors provide a very shallow description of RNA-seq results without publishing raw data or count tables. I believe that this is a necessary requirement in modern science.
Line 41: What does ‘multilineage’ mean in this context?
Line 48: what does ‘FCTC activation’ mean?
Line 49-52: ‘Major milestones during lizard blastema development can be grouped into four phases’ - it is incorrect to call the intact tail ‘resting phase’, as this state is not part of the regeneration process.
Line 74-76: clodronate induces apoptosis of osteoclasts, i.e. treatment with this agent suppresses their histolytic activity and all others. How can the authors judge that the pro-regenerative signal is separate from the histolytic signal? To prove this, histolysis should be switched off, but not osteoclasts killed.
Line 84: what does ‘stimulating blastema’ mean?
Line 142-164: The HCR procedure is not satisfactorily described. The reference to the Molecular Instruments protocol does not fulfil the requirements for reproducibility of the experiment, as the manufacturer is a third party and is not obliged to keep all versions of the protocols in the public domain. The protocol must be described exhaustively so that any reader can reproduce it. It should be indicated to which genes and how the probes were designed, what kind of hairpins with what fluorophores were used.
Line 185-187: I would ask the authors to provide an image from TapeStation with the results of RNA quality assessment, as after fixation in formalin, decalcification, HCR and flow cytometry, the level of RNA quality can be a challenge.
Line 198: reads were not subjected to trimming? This is unusual.
Line 200: for DESeq2, the version should be specified.
Line 209: ‘EdU’ instead of ‘Edu’.
Line 209-216: same remarks as for the HCR description.
Line 221 (and further in similar cases): ‘Previously described col3a1+ fibroblast populations’ - reference the source?
Line 221-222: the authors state that ‘col3a1+ fibroblast populations have been shown to expand to form the blastema’, but the images show that other col3a1- cells are present in the blastema. The blastema is formed with col3a1+ but not exclusively by them. This should be clarified in the text, as the current impression is that the blastema consists only of col3a1+ fibroblasts.
Line 225: what is [figure insert] for?
Line 230-242: the combination of colours cyan (col3a1+) and green (EdU+) for Figure 1 is unsuitable, as it is impossible to distinguish EdU+ nuclei from EdU- on the cyan background. Also, the resolution of the images (or lack of magnified inserts) does not allow us to determine where the EdU+ nuclei are. Therefore, I cannot consider the statement on lines 230-242 to be proven.
Line 266: «PBS» instead of «pbs».
Line 316 and further in similar cases: «DAPI» instead of «Dapi».
Line 342-343: This should be specified in the methods, not the results, as biological replicates describe the acquisition of material.
Line 347: DESeq2 instead of deseq2.
Line 348: is Figure 4C described here correctly?
Line 368-370: what is Z-score, how should the reader interpret it? Figure 4C lists all the genes that are differentially expressed at these stages of regeneration? I.e. are there 34 in total?
Line 385: is Figure 5C described here? Figure 5C does not list the col9a1 gene. Again, the heatmap shows all the genes that show differential expression?
Line 400: what does ‘Matrix associated genes are highly dysregulated’ mean? All genes, or just the few genes shown on the heatmap? Is the expression of other extracellular matrix proteins altered?
Line 525: this section describes a potential conflict of interest between authors. Please refer to the instructions for authors.
Overall, the main problem, in my opinion, is the poor discussion of the results and their integration in light of the data available today on a well-developed model such as vertebrate limb regeneration. The manuscript needs serious revision.
Round 2
Reviewer 2 Report
Comments and Suggestions for Authors
The reviewer's comments have been addressed. However, the text requires some editing, which should be provided by the journal (in some places there are missing spaces in the right places). In addition, although the authors' reply states that the raw data have been deposited in NCBI and the access number will be given after its receipt, such information is missing in the text. I request the editor's attention to ensure that these data are included in the final version.
Author Response
However, the text requires some editing, which should be provided by the journal (in some places there are missing spaces in the right places). In addition, although the authors' reply states that the raw data have been deposited in NCBI and the access number will be given after its receipt, such information is missing in the text. I request the editor's attention to ensure that these data are included in the final version.
We have adjusted the spacing and formatting as requested. We have also included the GSE number in the methods section of the text "Raw and normalized count files were uploaded to the NCBI Gene Expression Omnibus under GSE292905." 253-254
We appreciate your feedback!